# A Scalable Laplace Approximation for Neural Networks

**Hippolyt Ritter**[1][*]**, Aleksandar Botev**[1]**, David Barber**[1] [2]
[1]University College London [2]Alan Turing Institute

## Abstract

We leverage recent insights from second-order optimisation for neural networks to construct a Kronecker factored Laplace approximation to the posterior over the weights of a trained network. Our approximation requires no modification of the training procedure, enabling practitioners to estimate the uncertainty of their models currently used in production without having to retrain them. We extensively compare our method to using Dropout and a diagonal Laplace approximation for estimating the uncertainty of a network. We demonstrate that our Kronecker factored method leads to better uncertainty estimates on out-of-distribution data and is more robust to simple adversarial attacks. Our approach only requires calculating two square curvature factor matrices for each layer. Their size is equal to the respective square of the input and output size of the layer, making the method efficient both computationally and in terms of memory usage. We illustrate its scalability by applying it to a state-of-the-art convolutional network architecture.

## 1 Introduction

Neural networks are most commonly trained in a maximum a posteriori (MAP) setting, which only yields point estimates of the parameters, ignoring any uncertainty about them. This often leads to overconfident predictions, especially in regimes that are weakly covered by training data or far away from the data manifold. While the confidence of wrong predictions is usually irrelevant in a research context, it is essential that a Machine Learning algorithm knows when it does not know in the real world, as the consequences of mistakes can be fatal, be it when driving a car or diagnosing a disease.

The Bayesian framework of statistics provides a principled way for avoiding overconfidence in the parameters by treating them as unknown quantities and integrating over all possible values. Specifically, for the prediction of new data under a model, it fits a posterior distribution over the parameters given the training data and weighs the contribution of each setting of the parameters to the prediction by the probability of the data under those parameters times their prior probability. However, the posterior of neural networks is usually intractable due to their size and nonlinearity.

There has been previous interest in integrating neural networks into the Bayesian framework (MacKay, 1992; Hinton & Van Camp, 1993; Neal, 1993; Barber & Bishop, 1998), however these approaches were designed for small networks by current standards. Recent adaptations to architectures of modern scale rely on crude approximations of the posterior to become tractable. All of (Graves, 2011; Hernández-Lobato & Adams, 2015; Blundell et al., 2015) assume independence between the individual weights. While they achieve good results on small datasets, this strong restriction of the posterior is susceptible to underestimating the uncertainty, in particular when optimising the variational bound. The approach in (Gal & Ghahramani, 2016) requires the use of certain stochastic regularisers which are not commonly present in most recent architectures. Furthermore, it is not clear if the approximate posterior defined by these regularisers is a good fit to the true posterior.

Recent work on second-order optimisation of neural networks (Martens & Grosse, 2015; Botev et al., 2017) has demonstrated that the diagonal blocks of the curvature can be well approximated by a Kronecker product. We combine this insight with the idea of modelling the posterior over the weights as a Gaussian, using a Laplace approximation (MacKay, 1992) with Kronecker factored covariance matrices. This leads to a computationally efficient *matrix* normal posterior distribution

---

[*]Corresponding author: `j.ritter@cs.ucl.ac.uk`

(Gupta & Nagar, 1999) over the weights of every layer. Since the Laplace approximation is applied after training, our approach can be used to obtain uncertainty estimates from existing networks.

## 2 THE CURVATURE OF NEURAL NETWORKS

Our method is inspired by recent Kronecker factored approximations of the curvature of a neural network (Martens & Grosse, 2015; Botev et al., 2017) for optimisation and we give a high-level review of these in the following. While the two methods approximate the Gauss-Newton and Fisher matrix respectively, as they are guaranteed to be positive semi-definite (p.s.d.), we base all of our discussion on the Hessian in order to be as general as possible.

### 2.1 NEURAL NETWORK NOTATION

We denote a feedforward network as taking an input $a_0 = x$ and producing an output $h_L$. The intermediate representations for layers $\lambda = 1, ..., L$ are denoted as $h_\lambda = W_\lambda a_{\lambda-1}$ and $a_\lambda = f_\lambda(h_\lambda)$. We refer to $a_\lambda$ as the activations, and $h_\lambda$ as the (linear) pre-activations. The bias terms are absorbed into the $W_\lambda$ by appending a 1 to each $a_\lambda$. The network parameters are optimised w.r.t. an error function $E(y, h_L)$ for targets $y$. Most commonly used error functions, such as squared error and categorical cross-entropy, can be interpreted as exponential family negative log likelihoods $-\log p(y|h_L)$.

### 2.2 KRONECKER FACTORED SECOND-ORDER OPTIMISATION

Traditional second-order methods use either the Hessian matrix or a positive semi-definite approximation thereof to generate parameter updates of the form $\Delta = C^{-1}g$, where $C$ is the chosen curvature matrix and $g$ the gradient of the error function parameterised by the network. However, this curvature matrix is infeasbile to compute for modern neural networks as their number of parameters is often in the millions, rendering the size of $C$ of the order of several terabytes.

Recent work (Martens & Grosse, 2015; Botev et al., 2017) exploits that, for *a single data point*, the diagonal blocks of these curvature matrices are Kronecker factored:

$$H_\lambda = \frac{\partial^2 E}{\partial \operatorname{vec}(W_\lambda) \partial \operatorname{vec}(W_\lambda)} = \mathcal{Q}_\lambda \otimes \mathcal{H}_\lambda \tag{1}$$

where $H_\lambda$ is the Hessian w.r.t. the weights in layer $\lambda$. $\mathcal{Q}_\lambda = a_{\lambda-1} a_{\lambda-1}^\mathsf{T}$ denotes the covariance of the incoming activations $a_{\lambda-1}$ and $\mathcal{H}_\lambda = \frac{\partial^2 E}{\partial h_\lambda \partial h_\lambda}$ the pre-activation Hessian, i.e. the Hessian of the error w.r.t. the linear pre-activations $h_\lambda$ in a layer. We provide the derivation for this result as well as the recursion for calculating $\mathcal{H}$ in Appendix A.

The Kronecker factorisation holds two key advantages: the matrices that need be computed and stored are much smaller — if we assume all layers to be of dimensionality $D$, the two factors are each of size $D^2$, whereas the full Hessian for the weights of only one layer would have $D^4$ elements. Furthermore, the inverse of a Kronecker product is equal to the Kronecker product of the inverses, so it is only necessary to invert those two moderately sized matrices.

In order to maintain this structure over a minibatch of data, all Kronecker factored second-order methods make two core approximations: First, they only model the diagonal blocks corresponding to the weights of a layer, such that the curvature decomposes into $L$ independent matrices. Second, they assume $\mathcal{Q}_\lambda$ and $\mathcal{H}_\lambda$ to be independent. This is in order to maintain the Kronecker factorisation in expectation, i.e. $\mathbb{E}[\mathcal{Q}_\lambda \otimes \mathcal{H}_\lambda] \approx \mathbb{E}[\mathcal{Q}_\lambda] \otimes \mathbb{E}[\mathcal{H}_\lambda]$, since the expectation of a Kronecker product is not guaranteed to be Kronecker factored itself.

The main difference between the Kronecker factored second-order optimisers lies in how they efficiently approximate $\mathbb{E}[\mathcal{H}_\lambda]$. For exact calculation, it would be necessary to pass back an entire matrix per data point in a minibatch, which imposes infeasible memory and computational requirements. KFRA (Botev et al., 2017) simply passes back the expectation at every layer, while KFAC (Martens & Grosse, 2015) utilises the Fisher identity to only propagate a vector rather than a matrix, approximating the Kronecker factors with a stochastic rank-one matrix for each data point.

The diagonal blocks of the Hessian and Gauss-Newton matrix are equal for neural networks with piecewise linear activation functions (Botev et al., 2017), thus both methods can be used to directly approximate the diagonal blocks of the Hessian of such networks, as the Gauss-Newton and Fisher are equivalent for networks that parameterise an exponential family log likelihood.

# 3 A SCALABLE LAPLACE APPROXIMATION FOR NEURAL NETWORKS

## 3.1 THE LAPLACE APPROXIMATION

The standard Laplace approximation is obtained by taking the second-order Taylor expansion around a mode of a distribution. For a neural network, such a mode can be found using standard gradient-based methods. Specifically, if we approximate the log posterior over the weights of a network given some data $\mathcal{D}$ around a MAP estimate $\theta^*$, we obtain:

$$\log p(\theta|\mathcal{D}) \approx \log p(\theta^*|\mathcal{D}) - \frac{1}{2}(\theta - \theta^*)^\mathsf{T} \bar{H}(\theta - \theta^*) \tag{2}$$

where $\theta = [\text{vec}(W_1), ..., \text{vec}(W_L)]$ is the stacked vector of weights and $\bar{H} = \mathbb{E}[H]$ the average Hessian of the negative log posterior[1]. The first order term is missing because we expand the function around a maximum $\theta^*$, where the gradient is zero. If we exponentiate this equation, it is easy to notice that the right-hand side is of Gaussian functional form for $\theta$, thus we obtain a normal distribution by integrating over it. The posterior over the weights is then approximated as Gaussian:

$$\theta \sim \mathcal{N}(\theta^*, \bar{H}^{-1}) \tag{3}$$

assuming $\bar{H}$ is p.s.d. We can then approximate the posterior mean when predicting on unseen data $D^*$ by averaging the predictions of $T$ Monte Carlo samples $\theta^{(t)}$ from the approximate posterior:

$$p(\mathcal{D}^*|\mathcal{D}) = \int p(\mathcal{D}^*|\theta)p(\theta|\mathcal{D})d\theta \approx \frac{1}{T}\sum_{t=1}^{T} p(\mathcal{D}^*|\theta^{(t)}) \tag{4}$$

## 3.2 DIAGONAL LAPLACE APPROXIMATION

Unfortunately, it is not feasible to compute or invert the Hessian matrix w.r.t. all of the weights jointly. An approximation that is easy to compute in modern automatic differentiation frameworks is the diagonal of the Fisher matrix $F$, which is simply the expectation of the squared gradients:

$$H \approx \text{diag}(F) = \text{diag}(\mathbb{E}\left[\nabla_\theta \log p(y|x) \nabla_\theta \log p(y|x)^\mathsf{T}\right]) = \text{diag}(\mathbb{E}\left[(\nabla_\theta \log p(y|x))^2\right]) \tag{5}$$

where $\text{diag}$ extracts the diagonal of a matrix or turns a vector into a diagonal matrix. Such diagonal approximations to the curvature of a neural network have been used successfully for pruning the weights (LeCun et al., 1990) and, more recently, for transfer learning (Kirkpatrick et al., 2017).

This corresponds to modelling the weights with a Normal distribution with diagonal covariance:

$$\text{vec}(W_\lambda) \sim \mathcal{N}(\text{vec}(W_\lambda^*), \text{diag}(F_\lambda)^{-1}) \quad \text{for } \lambda = 1, \ldots, L \tag{6}$$

Unfortunately, even if the Taylor approximation is accurate, this will place significant probability mass in low probability areas of the true posterior if some weights exhibit high covariance.

---

[1] The average Hessian is typically scaled by the number of data points $N$. In order to keep the notation uncluttered, we develop our basic methods in terms of the average Hessian and discuss the scaling separately.

### 3.3 Kronecker Factored Laplace Approximation

So while it is desirable to model the covariance between the weights, some approximations are needed in order to remain computationally efficient. First, we assume the weights of the different layers to be independent. This corresponds to the block-diagonal approximation in KFAC and KFRA, which empirically preserves sufficient information about the curvature to obtain competitive optimisation performance. For our purposes this means that our posterior factorises over the layers.

As discussed above, the Hessian of the log-likelihood for a single datapoint is Kronecker factored, and we denote the two factor matrices as $H_\lambda = \mathcal{Q}_\lambda \otimes \mathcal{H}_\lambda$.[2] By further assuming independence between $\mathcal{Q}$ and $\mathcal{H}$ in all layers, we can approximate the expected Hessian of each layer as:

$$\mathbb{E}\left[H_\lambda\right] = \mathbb{E}\left[\mathcal{Q}_\lambda \otimes \mathcal{H}_\lambda\right] \approx \mathbb{E}\left[\mathcal{Q}_\lambda\right] \otimes \mathbb{E}\left[\mathcal{H}_\lambda\right] \tag{7}$$

Hence, the Hessian of every layer is Kronecker factored over an *entire dataset* and the Laplace approximation can be approximated by a product of Gaussians. Each Gaussian has a Kronecker factored covariance, corresponding to a *matrix* normal distribution (Gupta & Nagar, 1999), which considers the two Kronecker factors of the covariance to be the covariances of the rows and columns of a matrix. The two factors are much smaller than the full covariance and allow for significantly more efficient inversion and sampling (we review the matrix normal distribution in Appendix B).

Our resulting posterior for the weights in layer $\lambda$ is then:

$$W_\lambda \sim \mathcal{MN}(W_\lambda^*, \bar{\mathcal{Q}}_\lambda^{-1}, \bar{\mathcal{H}}_\lambda^{-1}) \tag{8}$$

In contrast to optimisation methods, we do not need to approximate $\mathbb{E}\left[\mathcal{H}_\lambda\right]$ as it is only calculated once. However, when it is possible to augment the data (e.g. randomised cropping of images), it may be advantageous. We provide a more detailed discussion of this in Appendix C.

### 3.4 Incorporating the Prior and Regularising the Curvature Factors

Just as the log posterior, the Hessian decomposes into a term depending on the data log likelihood and one on the prior. For the commonly used $L_2$-regularisation, corresponding to a Gaussian prior, the Hessian is equal to the precision of the prior times the identity matrix. We approximate this by adding a multiple of the identity to each of the Kronecker factors from the log likelihood:

$$H_\lambda = N\,\mathbb{E}\left[-\frac{\partial^2 \log p(\mathcal{D}|\theta)}{\partial \theta^2}\right] + \tau I \approx (\sqrt{N}\,\mathbb{E}\left[\mathcal{Q}_\lambda\right] + \sqrt{\tau}I) \otimes (\sqrt{N}\,\mathbb{E}\left[\mathcal{H}_\lambda\right] + \sqrt{\tau}I) \tag{9}$$

where $\tau$ is the precision of the Gaussian prior on the weights and $N$ the size of the dataset. However, we can also treat them as hyperparameters and optimise them w.r.t. the predictive performance on a validation set. We emphasise that this can be done without retraining the network, so it does not impose a large computational overhead and is trivial to parallelise.

Setting $N$ to a larger value than the size of the dataset can be interpreted as including duplicates of the data points as pseudo-observations. Adding a multiple of the uncertainty to the precision matrix decreases the uncertainty about each parameter. This has a regularising effect both on our approximation to the true Laplace, which may be overestimating the variance in certain directions due to ignoring the covariances between the layers, as well as the Laplace approximation itself, which may be placing probability mass in low probability areas of the true posterior.

## 4 Related Work

Most recent attempts to approximating the posterior of a neural network are based on formulating an approximate distribution to the posterior and optimising the variational lower bound w.r.t. its

---

[2]We assume a uniform prior for now, such that the Hessians of the posterior and the log likelihood are equal. We discuss how we incorporate a non-zero Hessian of a prior into the Kronecker factors in the next section.

parameters. (Graves, 2011; Blundell et al., 2015; Kingma et al., 2015) as well as the expectation propagation based approaches of (Hernández-Lobato & Adams, 2015) and (Ghosh et al., 2016) assume independence between the individual weights which, particularly when optimising the KL divergence, often lets the model underestimate the uncertainty about the weights. Gal & Ghahramani (2016) interpret Dropout to approximate the posterior with a mixture of delta functions, assuming independence between the columns. (Lakshminarayanan et al., 2016) suggest using an ensemble of networks for estimating the uncertainty.

Our work is a scalable approximation of (MacKay, 1992). Since the per-layer Hessian of a neural network is infeasible to compute, we suggest a factorisation of the covariance into a Kronecker product, leading to a more efficient *matrix* normal distribution. The posterior that we obtain is reminiscent of (Louizos & Welling, 2016) and (Sun et al., 2017), who optimise the parameters of a matrix normal distribution as their weights, which requires a modification of the training procedure.

## 5 EXPERIMENTS

Since the Laplace approximation is a method for *predicting* in a Bayesian manner and not for training, we focus on comparing to uncertainty estimates obtained from Dropout (Gal & Ghahramani, 2016). The trained networks will be identical, but the prediction methods will differ. We also compare to a diagonal Laplace approximation to highlight the benefit from modelling the covariances between the weights. All experiments are implemented using Theano (Theano Development Team, 2016) and Lasagne (Dieleman et al., 2015).[3]

### 5.1 TOY REGRESSION DATASET

As a first experiment, we visualise the uncertainty obtained from the Laplace approximations on a toy regression dataset, similar to (Hernández-Lobato & Adams, 2015). We create a dataset of 20 uniformly distributed points $x \sim \mathcal{U}(-4, 4)$ and sample $y \sim \mathcal{N}(x^3, 3^2)$. In contrast to (Hernández-Lobato & Adams, 2015), we use a two-layer network with seven units per layer rather than one layer with 100 units. This is because both the input and output are one-dimensional, hence the weight matrices are vectors and the matrix normal distribution reduces to a multivariate normal distribution. Furthermore, the Laplace approximation is sensitive to the ratio of the number of data points to parameters, and we want to visualise it both with and without hyperparameter tuning.

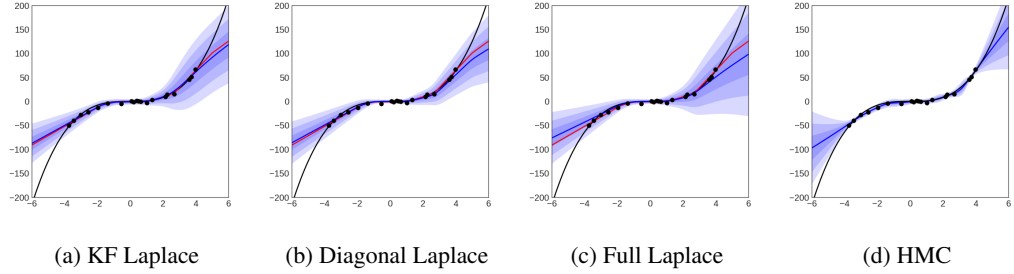

| (a) KF Laplace | (b) Diagonal Laplace | (c) Full Laplace | (d) HMC |

Figure 1: Toy regression uncertainty. Black dots are data points, the black line shows the noiseless function. The red line shows the deterministic prediction of the network, the blue line the mean output. Each shade of blue visualises one additional standard deviation. Best viewed on screen.

Fig. 1 shows the uncertainty obtained from the Kronecker factored and diagonal Laplace approximation applied to the same network, as well as from a full Laplace approximation and $50,000$ HMC (Neal, 1993) samples. The latter two methods are feasible only for such a small model and dataset. For the diagonal and full Laplace approximation we use the Fisher identity and draw one sample per data point. We set the hyperparameters of the Laplace approximations (see Section 3.4) using a grid search over the likelihood of 20 validation points that are sampled the same way as the training set.

---

[3]We make our fork available at: `https://github.com/BB-UCL/Lasagne`

The regularised Laplace approximations all give an overall good fit to the HMC predictive posterior. Their uncertainty is slightly higher close to the training data and increases more slowly away from the data than that of the HMC posterior. The diagonal and full Laplace approximation require stronger regularisation than our Kronecker factored one, as they have higher uncertainty when not regularised. In particular the full Laplace approximation vastly overestimates the uncertainty without additional regularisation, leading to a bad predictive mean (see Appendix E for the corresponding figures), as the Hessian of the log likelihood is underdetermined. This is commonly the case in deep learning, as the number of parameters is typically much larger than the number of data points. Hence restricting the structure of the covariance is not only a computational necessity for most architectures, but also allows for more precise estimation of the approximate covariance.

## 5.2 Out-of-Distribution Uncertainty

For a more realistic test, similar to (Louizos & Welling, 2017), we assess the uncertainty of the predictions when classifying data from a different distribution than the training data. For this we train a network with two layers of $1024$ hidden units and ReLU transfer functions to classify MNIST digits. We use a learning rate of $10^{-2}$ and momentum of $0.9$ for $250$ epochs. We apply Dropout with $p=0.5$ after each inner layer, as our chief interest is to compare against its uncertainty estimates. We further use $L_2$-regularisation with a factor of $10^{-2}$ and randomly binarise the images during training according to their pixel intensities and draw $1,000$ such samples per datapoint for estimating the curvature factors. We use this network to classify the images in the notMNIST dataset[4], which contains $28 \times 28$ grey-scale images of the letters 'A' to 'J' from various computer fonts, i.e. not digits. An ideal classifier would make uniform predictions over its classes.

We compare the uncertainty obtained by predicting the digit class of the notMNIST images using 1. a deterministic forward pass through the Dropout trained network, 2. by sampling different Dropout masks and averaging the predictions, and by sampling different weight matrices from 3. the matrix normal distribution obtained from our Kronecker factored Laplace approximation as well as 4. the diagonal one. As an additional baseline similar to (Blundell et al., 2015; Graves, 2011), we compare to a network with identical architecture with a fully factorised Gaussian (FFG) approximate posterior on the weights and a standard normal prior. We train the model on the variational lower bound using the reparametrisation trick (Kingma & Welling, 2013). We use 100 samples for the stochastic forward passes and optimise the hyperparameters of the Laplace approximations w.r.t. the cross-entropy on the validation set of MNIST.

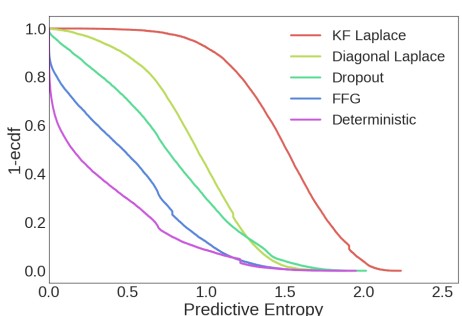

Figure 2: Predictive entropy on notMNIST obtained from different methods for the forward pass on a network trained on MNIST.

We measure the uncertainty of the different methods as the entropy of the predictive distribution, which has a minimal value of 0 when a single class is predicted with certainty and a maximum of about 2.3 for uniform predictions. Fig. 2 shows the inverse empirical cumulative distribution of the entropy values obtained from the four methods. Consistent with the results in (Gal & Ghahramani, 2016), averaging the probabilities of multiple passes through the network yields predictions with higher uncertainty than a deterministic pass that approximates the geometric average (Srivastava et al., 2014). However, there still are some images that are predicted to be a digit with certainty. Our Kronecker factored Laplace approximation makes hardly any predictions with absolute certainty and assigns high uncertainty to most of the letters as desired. The diagonal Laplace approximation required stronger regularisation towards predicting deterministically, yet it performs similarly to Dropout. As shown in Table 1, however, the network makes predictions on the test set of MNIST

---

[4]From: `http://yaroslavvb.blogspot.nl/2011/09/notmnist-dataset.html`

with similar accuracy to the deterministic forward pass and MC Dropout when using our approximation. The variational factorised Gaussian posterior has low uncertainty as expected.

## 5.3 Adversarial Examples

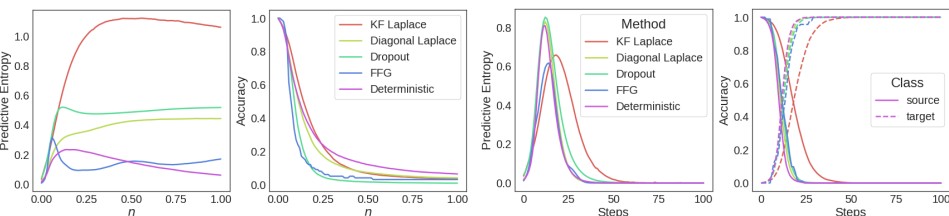

Figure 3: Untargeted adversarial attack.                Figure 4: Targeted adversarial attack.

To further test the robustness of our prediction method close to the data distribution, we perform an adversarial attack on a neural network. As first demonstrated in (Szegedy et al., 2013), neural networks are prone to being fooled by gradient-based changes to their inputs. Li & Gal (2017) suggest, and provide empirical support, that Bayesian models may be more robust to such attacks, since they implicitly form an infinitely large ensemble by integrating over the model parameters. For our experiments, we use the fully connected net trained on MNIST from the previous section and compare the sensitivity of the different prediction methods for two kinds of adversarial attacks.

First, we use the untargeted Fast Gradient Sign method $x_{adv} = x - \eta \operatorname{sgn}(\nabla_x \max_y \log p^{(M)}(y|x))$ suggested in (Goodfellow et al., 2014), which takes the gradient of the class predicted with maximal probability by method $M$ w.r.t. the input $x$ and reduces this probability with varying step size $\eta$. This step size is rescaled by the difference between the maximal and minimal value per dimension in the dataset. It is to be expected that this method generates examples away from the data manifold, as there is no clear subset of the data that corresponds to e.g. "not ones".

Fig. 3 shows the average predictive uncertainty and the accuracy on the original class on the MNIST test set as the step size $\eta$ increases. The Kronecker factored Laplace approximation achieves significantly higher uncertainty than any other prediction method as the images move away from the data. Both the diagonal and the Kronecker factored Laplace maintain higher accuracy than MC Dropout on their original predictions. Interestingly, the deterministic forward pass appears to be most robust in terms of accuracy, however it has much smaller uncertainty on the predictions it makes and will confidently predict a false class for most images, whereas the other methods are more uncertain.

Furthermore, we perform a targeted attack that attempts to force the network to predict a specific class, in our case '0' following (Li & Gal, 2017). Hence, for each method, we exclude all data points in the test set that are already predicted as '0'. The updates are of similar form to the untargeted attack, however they increase the probability of the pre-specified class $y$ rather than decreasing the current maximum as $x_y^{(t+1)} = x_y^{(t)} + \eta \operatorname{sgn}(\nabla_x \log p^{(M)}(y|x_y^{(t)}))$, where $x_y^{(0)} = x$.

We use a step size of $\eta = 10^{-2}$ for the targeted attack. The uncertainty and accuracy on the original and target class are shown in Fig. 4. Here, the Kronecker factored Laplace approximation has slightly smaller uncertainty at its peak in comparison to the other methods, however it appears to be much more robust. It only misclassifies over 50% of the images after about 20 steps, whereas for the other methods this is the case after roughly 10 steps and reaches 100% accuracy on the target class after almost 50 updates, whereas the other methods are fooled on all images after about 25 steps.

In conjunction with the experiment on notMNIST, it appears that the Laplace approximation achieves higher uncertainty than Dropout away from the data, as in the untargeted attack. In the targeted attack it exhibits smaller uncertainty than Dropout, yet it is more robust to having its prediction changed. The diagonal Laplace approximation again performs similarly to Dropout.

## 5.4 Uncertainty on Misclassifications

To highlight the scalability of our method, we apply it to a state-of-the-art convolutional network architecture. Recently, deep residual networks (He et al., 2016a;b) have been the most successful

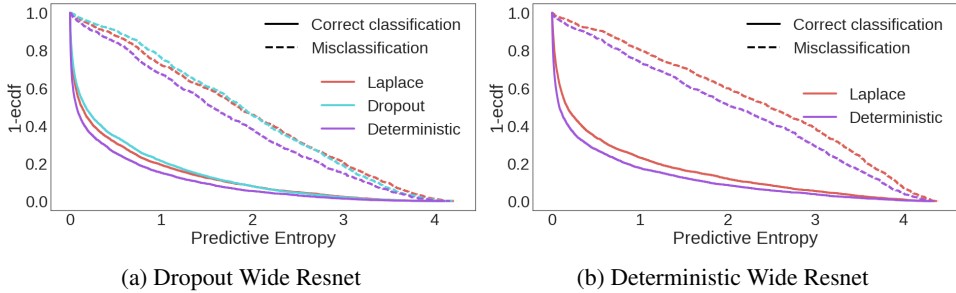

(a) Dropout Wide Resnet        (b) Deterministic Wide Resnet

Figure 5: Inverse ecdf of the predictive entropy from Wide Residual Networks trained with and without Dropout on CIFAR100. For misclassifications, curves on top corresponding to higher uncertainty are desirable, and curves on the bottom for correct classifications.

ones among those. As demonstrated in (Grosse & Martens, 2016), Kronecker factored curvature methods are applicable to convolutional layers by interpreting them as matrix-matrix multiplications.

We compare our uncertainty estimates on wide residual networks (Zagoruyko & Komodakis, 2016), a recent variation that achieved competitive performance on CIFAR100 (Krizhevsky & Hinton, 2009) while, in contrast to most other residual architectures, including Dropout at specific points. While this does not correspond to using Dropout in the Bayesian sense (Gal & Ghahramani, 2015), it allows us to at least compare our method to the uncertainty estimates obtained from Dropout.

We note that it is straightforward to incorporate batch normalisation (Ioffe & Szegedy, 2015) into the curvature backpropagation algorithms, so we apply a standard Laplace approximation to its parameters as well. We are not aware of any interpretation of Dropout as performing Bayesian inference on the parameters of batch normalisation. Further implementation details are in Appendix G.

Again, the accuracy of the prediction methods is comparable (see Table 2 in Appendix F). For calculating the curvature factors, we draw $5,000$ samples per image using the same data augmentation as during training, effectively increasing the dataset size to $2.5{\times}10^8$. The diagonal approximation had to be regularised to the extent of becoming deterministic, so we omit it from the results.

In Fig. 5 we compare the distribution of the predictive uncertainty on the test set.[5] We distinguish between the uncertainty on correct and incorrect classifications, as the mistakes of a system used in practice may be less severe if the network can at least indicate that it is uncertain. Thus, high uncertainty on misclassifications and low uncertainty on correct ones would be desirable, such that a system could return control to a human expert when it can not make a confident decision. In general, the network tends to be more uncertain on its misclassifcations than its correct ones regardless of whether it was trained with or without Dropout and of the method used for prediction. Both Dropout and the Laplace approximation similarly increase the uncertainty in the predictions, however this is irrespective of the correctness of the classification. Yet, our experiments show that the Kronecker factored Laplace approximation can be scaled to modern convolutional networks and maintain good classification accuracy while having similar uncertainty about the predictions as Dropout.

We had to use much stronger regularisation for the Laplace approximation on the wide residual network, possibly because the block-diagonal approximation becomes more inaccurate on deep networks, possibly because the number of parameters is much higher relative to the number of data. It would be interesting to see how the Laplace approximations behaves on a much larger dataset like ImageNet for similarly sized networks, where we have a better ratio of data to parameters and curvature directions. However, even on a relatively small dataset like CIFAR we did not have to regularise the Laplace approximation to the degree of the posterior becoming deterministic.

---

[5]We use the first $5,000$ images as a validation set to tune the hyperparameters of our Laplace approximation and the final $5,000$ ones for evaluating the predictive uncertainty on all methods.

## 6 CONCLUSION

We presented a scalable approximation to the Laplace approximation for the posterior of a neural network and provided experimental results suggesting that the uncertainty estimates are on par with current alternatives like Dropout, if not better. It enables practitioners to obtain principled uncertainty estimates from their models, even if they were trained in a maximum likelihood/MAP setting.

There are many possible extensions to this work. One would be to automatically determine the scale and regularisation hyperparameters of the Kronecker factored Laplace approximation using the model evidence similar to how (MacKay, 1992) interpolates between the data log likelihood and the width of the prior. The model evidence could further be used to perform Bayesian model averaging on ensembles of neural networks, potentially improving their generalisation ability and uncertainty estimates. A challenging application would be active learning, where only little data is available relative to the number of curvature directions that need to be estimated.

### ACKNOWLEDGEMENTS

This work was supported by the Alan Turing Institute under the EPSRC grant EP/N510129/1. We thank the anonymous reviewers for their feedback and Harshil Shah for his comments on an earlier draft of this paper.

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

# Appendices

## A   DERIVATION OF THE ACTIVATION HESSIAN RECURSION

Here, we provide the basic derivation of the factorisation of the diagonal blocks of the Hessian in Eq. 1 and the recursive formula for calculating $\mathcal{H}$ as presented in (Botev et al., 2017).

The Hessian of a neural network with parameters $\theta$ as defined in the main text has elements:

$$[H]_{ij} = \frac{\partial^2}{\partial \theta_i \partial \theta_j} E(\theta) \tag{10}$$

For a given layer $\lambda$, the gradient w.r.t. a weight $W_{a,b}^\lambda$ is:

$$\frac{\partial E}{\partial W_{a,b}^\lambda} = \sum_i \frac{\partial h_i^\lambda}{\partial W_{a,b}^\lambda} \frac{\partial E}{\partial h_i^\lambda} = a_b^{\lambda-1} \frac{\partial E}{\partial h_a^\lambda} \tag{11}$$

Keeping $\lambda$ fixed and differentiating again, we find that the per-sample Hessian of that layer is:

$$[H_\lambda]_{(a,b),(c,d)} \equiv \frac{\partial^2 E}{\partial W_{a,b}^\lambda \partial W_{c,d}^\lambda} = a_b^{\lambda-1} a_d^{\lambda-1} [\mathcal{H}_\lambda]_{a,c} \tag{12}$$

where

$$[\mathcal{H}_\lambda]_{a,b} = \frac{\partial^2 E}{\partial h_a^\lambda \partial h_b^\lambda} \tag{13}$$

is the pre-activation Hessian.

We can reexpress this in matrix notation as a Kronecker product as in Eq. 1:

$$H_\lambda = \frac{\partial^2 E}{\partial \operatorname{vec}(W_\lambda) \partial \operatorname{vec}(W_\lambda)} = \left( a_{\lambda-1} a_{\lambda-1}^\mathsf{T} \right) \otimes \mathcal{H}_\lambda \tag{14}$$

The pre-activation Hessian can be calculated recursively as:

$$\mathcal{H}_\lambda = B_\lambda W_{\lambda+1}^\mathsf{T} \mathcal{H}_{\lambda+1} W_{\lambda+1} B_\lambda + D_\lambda \tag{15}$$

where the diagonal matrices $B$ and $D$ are defined as:

$$B_\lambda = \operatorname{diag}\left( \mathbf{f}_\lambda'(h_\lambda) \right) \tag{16}$$

$$D_\lambda = \operatorname{diag}\left( \mathbf{f}_\lambda''(h_\lambda) \frac{\partial E}{\partial a_\lambda} \right) \tag{17}$$

$\mathbf{f}'$ and $\mathbf{f}''$ denote the first and second derivative of the transfer function. The recursion is initialised with the Hessian of the error w.r.t. the linear network outputs.

For further details and on how to calculate the diagonal blocks of the Gauss-Newton and Fisher matrix, we refer the reader to (Botev et al., 2017) and (Martens & Grosse, 2015).

## B    MATRIX NORMAL DISTRIBUTION

The matrix normal distribution (Gupta & Nagar, 1999) is a multivariate distribution over an entire matrix of shape $n \times p$ rather than just a vector. In contrast to the multivariate normal distribution, it is parameterised by two p.s.d. covariance matrices, $U : n \times n$ and $V : p \times p$, which indicate the covariance of the rows and columns respectively. In addition it has a mean matrix $M : n \times p$.

A vectorised sample from a matrix normal distribution $X \sim \mathcal{MN}(M, U, V)$ corresponds to a sample from a normal distribution $\text{vec}(X) \sim \mathcal{N}(\text{vec}(M), U \otimes V)$. However, samples can be drawn more efficiently as $X = M + AZB$ with $Z \sim \mathcal{MN}(0, I, I)$, and $AA^\mathsf{T} = U$ and $B^\mathsf{T}B = V$. The sample $Z$ corresponds to a sample from a normal distribution of length $np$ that has been reshaped to a $n \times p$ matrix. This is more efficient in the sense that we only need to calculate two matrix-matrix products of small matrices, rather than a matrix-vector product with one big one.

## C    APPROXIMATION OF THE EXPECTED ACTIVATION HESSIAN

While the square root of $\mathcal{Q}_\lambda$ is calculated during the forward pass on all layers, $\mathcal{H}$ requires an additional backward pass. Strictly speaking, it is not essential to approximate $\mathbb{E}[\mathcal{H}]$ for the Kronecker factored Laplace approximation, as in contrast to optimisation procedures the curvature only needs to be calculated once and is thus not time critical. For datasets of the scale of ImageNet and the networks used for such datasets, it would still be impractically slow to perform the calculation for every data point individually. Furthermore, as most datasets are augmented during training, e.g. random cropping or reflections of images, the curvature of the network can be estimated using the same augmentations, effectively increasing the size of the dataset by orders of magnitude. Thus, we make use of the minibatch approximation in our experiments — as we make use of data augmentation — in order to demonstrate its practical applicability.

We note that $\mathbb{E}[\mathcal{H}]$ can be calculated exactly by running KFRA (Botev et al., 2017) with a minibatch-size of one, and then averaging the results. KFAC (Martens & Grosse, 2015), in contrast, stochastically approximates the Fisher matrix, so even when run for every datapoint separately, it cannot calculate the curvature factor exactly.

In the following, we also show figures for the adversarial experiments in which we calculate the curvature per datapoint and without data augmentation:

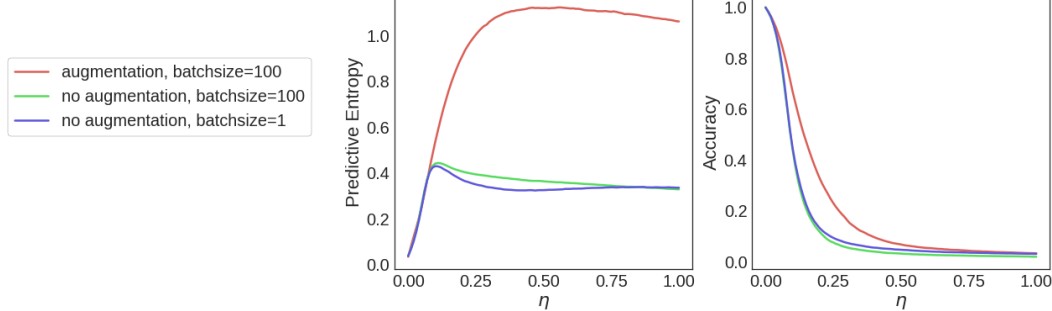

Figure 6: Untargeted adversarial attack for Kronecker factored Laplace approximation with the curvature calculated with and without data augmentation/approximating the activation Hessian.

Fig. 6 and Fig. 7 show how the Laplace approximation with the curvature estimated from 1000 randomly sampled binary MNIST images and the activation Hessian calculated with a minibatch size of 100 performs in comparison to the curvature factor being calculated without any data augmentation with a batch size of 100 or exactly. We note that without data augmentation we had to use much stronger regularisation of the curvature factors, in particular we had to add a non-negligible multiple of the identity to the factors, whereas with data augmentation it was only needed to ensure that the matrices are invertible. The Kronecker factored Laplace approximation reaches particularly high uncertainty on the untargeted adversarial attack and is most robust on the targeted attack when using data augmentation, suggesting that it is particularly well suited for large datasets and ones

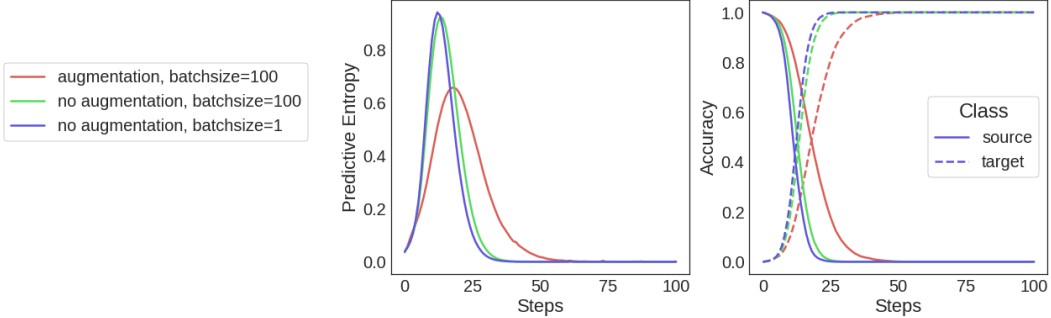

Figure 7: Targeted adversarial attack for Kronecker factored Laplace approximation with the curvature calculated with and without data augmentation/approximating the activation Hessian.

where some form of data augmentation can be applied. The difference between approximating the activation Hessian over a minibatch and calculating it exactly appears to be negligible.

## D    MEMORY AND COMPUTATIONAL REQUIREMENTS

If we denote the dimensionality of the input to layer $\lambda$ as $D_{\lambda-1}$ and its output as $D_\lambda$, the curvature factors correspond to the two precision matrices with $\frac{D_{\lambda-1}(D_{\lambda-1}+1)}{2}$ and $\frac{D_\lambda(D_\lambda+1)}{2}$ 'parameters' to estimate, since they are symmetric. So across a network, the number of curvature directions that we are estimating grows linearly in the number of layers and quadratically in the dimension of the layers, i.e. the number of columns of the weight matrices. The size of the full Hessian, on the other hand, grows quadratically in the number of layers and with the fourth power in the dimensionality of the layers (assuming they are all the same size).

Once the curvature factors are calculated, which only needs to be done once, we use their Cholesky decomposition to solve two triangular linear systems when sampling weights from the matrix normal distribution. We use the same weight samples for each minibatch, i.e. we do not sample a weight matrix per datapoint. This is for computational efficiency and does not change the expectation.

One possibility to save computation time would be to sample a fixed set of weight matrices from the approximate posterior — in order to avoid solving the linear system on every forward pass — and treat the networks that they define as an ensemble. The individual ensemble members can be evaluated in parallel and their outputs averaged, which can be done with a small overhead over evaluating a single network given sufficient compute resources. A further speed up can be achieved by distilling the predictive distributions of the Laplace network into a smaller, deterministic feedforward network as successfully demonstrated in (Balan et al., 2015) for posterior samples using HMC.

## E    COMPLEMENTARY FIGURES FOR THE TOY DATASET

Fig. 8 shows the different Laplace approximations (Kronecker factored, diagonal, full) from the main text without any hyperparameter tuning. The figure of the uncertainty obtained from samples using HMC is repeated. Note that the scale is larger than in the main text due to the high uncertainty of the Laplace approximations.

The Laplace approximations are increasingly uncertain away from the data, as the true posterior estimated from HMC samples, however they all overestimate the uncertainty without regularisation. This is easy to fix by optimising the hyperparameters on a validation set as discussed in the main text, resulting in posterior uncertainty much more similar to the true posterior. As previously discussed in (Botev et al., 2017), the Hessian of a neural network is usually underdetermined as the number of data points is much smaller than the number of parameters — in our case we have 20 data points to estimate a $78 \times 78$ precision matrix. This leads to the full Laplace approximation vastly overestimating the uncertainty and a bad predictive mean. Both the Kronecker factored and the diagonal approximation exhibit smaller variance than the full Laplace approximation as they restrict the structure of the precision matrix. Consistently with the other experiments, we find the diagonal

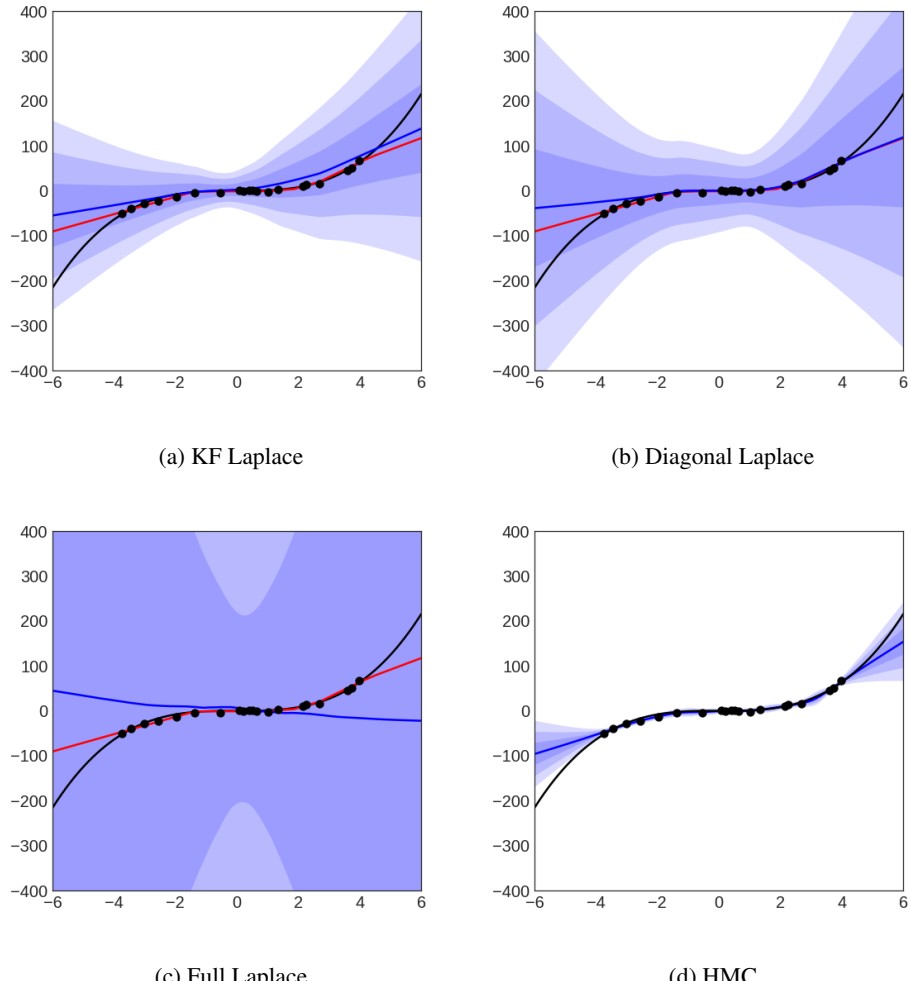

(a) KF Laplace

(b) Diagonal Laplace

(c) Full Laplace

(d) HMC

Figure 8: Toy regression uncertainty. Black dots are data points, the black line shows the underlying noiseless function. The red line shows the deterministic prediction of the trained network, the blue line the mean output. Each shade of blue visualises one additional standard deviation.

Laplace approximation to place more mass in low probability areas of the posterior than the Kronecker factored approximation, resulting in higher variance on the regression problem. This leads to a need for greater regularisation of the diagonal approximation to obtain acceptable predictive performance, and underestimating the uncertainty.

## F   PREDICTION ACCURACY

This section shows the accuracy values obtained from the different predictions methods on the feed-forward networks for MNIST and the wide residual network for CIFAR100. The results for MNIST are shown in Table 1 and the results for CIFAR in Table 2.

In all cases, neither MC Dropout nor the Laplace approximation significantly change the classification accuracy of the network in comparison to a deterministic forward pass.

Table 1: Test accuracy of the feedforward network trained on MNIST

| Prediction Method | Accuracy |
|---|---|
| FFG | 98.88% |
| Deterministic | 98.86% |
| MC Dropout | 98.85% |
| Diagonal Laplace | 98.85% |
| **KF Laplace** | 98.80% |

Table 2: Accuracy on the final $5,000$ CIFAR100 test images for a wide residual network trained with and without Dropout.

| | Accuracy | |
|---|---|---|
| Prediction Method | Dropout | Deterministic |
| Deterministic | 79.12% | 79.18% |
| MC Dropout | 79.20% | - |
| **KF Laplace** | 79.10% | 79.36% |

# G    IMPLEMENTATION DETAILS FOR RESIDUAL NETWORKS

Our wide residual network has $n=3$ block repetitions and a width factor of $k=8$ on CIFAR100 with and without Dropout using hyperparameters taken from (Zagoruyko & Komodakis, 2016): the network parameters are trained on a cross-entropy loss using Nesterov momentum with an initial learning rate of $0.1$ and momentum of $0.9$ for $200$ epochs with a minibatch size of $128$. We decay the learning rate every $50$ epochs by a factor of $0.2$, which is slightly different to the schedule used in (Zagoruyko & Komodakis, 2016) (they decay after $60$, $120$ and $160$ epochs). As the original authors, we use $L_2$-regularisation with a factor of $5\times10^{-4}$.

We make one small modification to the architecture: instead of downsampling with $1\times1$ convolutions with stride $2$, we use $2\times2$ convolutions. This is due to Theano not supporting the transformation of images into the patches extracted by a convolution for $1\times1$ convolutions with stride greater than $1$, which we require for our curvature backpropagation through convolutions.

We apply a standard Laplace approximation to the batch normalisation parameters — a Kronecker factorisation is not needed, since the parameters are one-dimensional. When calculating the curvature factors, we use the moving averages for the per-layer means and standard deviations obtained after training, in order to maintain independence between the data points in a minibatch.

We need to make a further approximation to the ones discussed in Section 2.2 when backpropagating the curvature for residual networks. The residual blocks compute a function of the form $res(x) = x + f_\phi(x)$, where $f_\phi$ typically is a sequence of convolutions, batch normalisation and elementwise nonlinearities. This means that we would need to pass back two curvature matrices, one for each summand. However, this would double the number of backpropagated matrices for each residual connection, hence the computation time/memory requirements would grow exponentially in the number of residual blocks. Therefore, we simply add the curvature matrices after each residual connection.

