# OpenReview forum: "A Scalable Laplace Approximation for Neural Networks"
_ICLR.cc/2018/Conference — Accept (Poster)_

### Official Review · AnonReviewer1 · 2017-11-26
**Novel idea, well-written, and supported with extensive experiments.**

**Rating:** 9
**Confidence:** 4

**Review:**

This paper proposes a novel scalable method for incorporating uncertainty estimate in neural networks, in addition to existing methods using, for example, variational inference and expectation propagation. The novelty is in extending the Laplace approximation introduced in MacKay (1992) using a Kronecker-factor approximation of the Hessian. The paper is well written and easy to follow. It provides extensive references to related works, and supports its claims with convincing experiments from different domains.

Pros:
-A novel method in an important and interesting direction.
-It is a prediction method, so can be applied on existing trained neural networks (however, see the first con).
-Well-written with high clarity.
-Extensive and convincing experiments.

Cons:
-Although it is a predictive method, it's still worth discussing how this method relates to training. For example, I suspect it works better when the model is trained with second-order method, as the resulting Taylor approximation (eq. 2) of the log-likelihood function might have higher quality when both terms are explicitly used in optimisation.
-The difference between using KFAC and KFRA is unclear, or should be better explained if they are identical in this context. Botev et al. 2017 reports they are slightly different in approximating the Gaussian Newton matrix.
-Acronyms, even well-known, are better defined before using (e.g., EP, PSD).
-Need more details of the optimisation method used in experiments, especially the last one.

---

> ### Author Response · Authors · 2018-01-05
> **Response to Reviewer 3**
>
> Thank you very much for your positive review, we have updated the manuscript to introduce all acronyms before using them and added details regarding the hyperparameters of the last experiment to the appendix.
>
> Regarding how the optimisation method affects the Laplace approximation is a question that we believe is closely related to how the optimisation method affects generalisation. We therefore decided to simply go with an optimiser that is commonly used in practice to make our results relevant to those who might use our method, however we are definitely open to adding an empirical comparison with different optimisation methods to a camera-ready version of the paper. Answering this question in full generality seems like a very interesting, but challenging open research problem.

---

### Official Review · AnonReviewer2 · 2017-11-26
**Good paper but what about novelty?**

**Rating:** 6
**Confidence:** 4

**Review:**

This paper proposes a Laplace approximation to approximate the posterior distribution over the parameters of deep networks.

The idea is interesting and the realization of the paper is good. The idea builds upon previous work in scalable Gauss-Newton methods for optimization in deep networks, notably Botev et al., ICML 2017. In this respect, I think that the novelty in the current submission is limited, as the approximation is essentially what proposed in Botev et al., ICML 2017.  The Laplace approximation requires the Hessian of the posterior, so techniques developed for Gauss-Newton optimization can straightforwardly be applied to construct Laplace approximations.

Having said that, the experimental evaluation is quite interesting and in-depth. I think it would have been interesting to report comparisons with factorized variational inference (Graves, 2011) as it is a fairly standard and widely adopted in Bayesian deep learning. This would have been an interesting way to support the claims on the poor approximation offered by standard variational inference.

I believe that the independence assumption across layers is a limiting factor of the proposed approximation strategy. Intuitively, changes in the weights in a given layer should affect the weights in other layers, so I would expect the posterior distribution over all the weights to reflect this through correlations across layers. I wonder how these results can be generalized to relax the independence assumption.

---

> ### Author Response · Authors · 2018-01-05
> **Response to Reviewer 2**
>
> Thank you very much for your comments and your review. We will address a few specific points that you raised in the following:
>
>
> >In this respect, I think that the novelty in the current submission is limited, as the approximation is essentially what proposed in Botev et al., ICML 2017.  The Laplace approximation requires the Hessian of the posterior, so techniques developed for Gauss-Newton optimization can straightforwardly be applied to construct Laplace approximations.
>
> We fully agree that, from a techincal perspective, the approximation to the Hessian is not new and that once the two Kronecker factors are calculated it is relatively straightforward (in terms of implementation) to calculate the approximate predicive mean for a network. However, we do think that introducing these ideas from the optimisation literature to the Bayesian deep learning community, demonstrating how the Laplace approximation can be scaled to neural networks, is indeed a novel and valuable contribution (since the diagonal approximation is not sufficient as shown in our experiments and the full approximation is not feasible). The Laplace approximation fundamentelly differs from the currently popular variational inference approaches in not requiring a modification to the training procedure, which is extremely useful for practictioners as they can simply apply it to their exisiting networks/do not need to do a full new hyperparameter search for optimising the parameters of an approximate posterior.
>
>
> >I think it would have been interesting to report comparisons with factorized variational inference (Graves, 2011) as it is a fairly standard and widely adopted in Bayesian deep learning. This would have been an interesting way to support the claims on the poor approximation offered by standard variational inference.
>
> We have added this baseline to the 2nd and 3rd experiment, as this was also requested by Reviewer 1 (our original aim was to have a "clean" comparison that is independent of the optimisation objective/procedure by focusing on different prediction methods for an identical network).
>
>
> >I believe that the independence assumption across layers is a limiting factor of the proposed approximation strategy. Intuitively, changes in the weights in a given layer should affect the weights in other layers, so I would expect the posterior distribution over all the weights to reflect this through correlations across layers. I wonder how these results can be generalized to relax the independence assumption.
>
> Thank you for this suggestion. Indeed, the layerwise blocks of the Fisher and Gauss-Newton are all Kronecker factored, so it should be possible to include the covariance of e.g. neighbouring layers in a computationally efficient way. In their work on KFAC, Martens & Grosse investigated such a tri-diagonal block approximation of the Fisher, however this only gave a minor improvment in performance over the block-diagonal approximation. Yet, since optimisation is a lot more time-critical, this could be worth investigating in the future for the Laplace approximation.

---

> > ### Comment · AnonReviewer2 · 2018-01-12
> > **Response to rebuttal**
> >
> > Many thanks for writing your rebuttal and for adding experiments on variational inference with fully factorized posterior. I believe that these comparisons add value to the proposal, given that the proposed approach achieves better performance. I'm keen to raise my score due to that, although I still think that the novelty can be an issue.

---

### Official Review · AnonReviewer3 · 2017-11-26
**Competent work building on recent literature. Mixed results and perhaps not currently fulfilling its full potential as a paper.**

**Rating:** 6
**Confidence:** 4

**Review:**

This paper uses recent progress in the understanding and approximation of curvature matrices in neural networks to revisit a venerable area- that of Laplace approximations to neural network posteriors. The Laplace method requires two stages - 1) obtaining a point estimate of the parameters followed by 2) estimation of the curvature. Since 1) is close to common practice it raises the appealing possibility of adding 2) after the fact, although the prior may be difficult to interpret in this case. A pitfall is that the method needs the point estimate to fall in a locally quadratic bowl or to add regularisation to make this true. The necessary amount of regularisation can be large as reported in section 5.4.

The paper is generally well written. In particular the mathematical exposition attains good clarity. Much of the mathematical treatment of the curvature was already discussed by Martens and Grosse and Botev et al in previous works. The paper is generally well referenced.

Given the complexity of the method, I think it would have helped to submit the code in anonymized form at this point.There are also some experiments not there that would improve the contribution. Figure 1 should include a comparison to Hamiltonian Monte Carlo and the full Laplace approximation (It is not sufficient to point to experiments in Hernandez-Lobato and Adams 2015 with a different model/prior). The size of model and data would not be prohibitive for either of these methods in this instance. All that figure 1 shows at the moment is that the proposed approximation has smaller predictive variance than the fully diagonal variant of the method.

It would be interesting (but perhaps not essential) to compare the Laplace approximation to other scalable methods from the literature such as that of Louizos and Welling 2016 which uses also used matrix normal distributions. It is good that the paper includes a modern architecture with a more challenging dataset. It is a shame the method does not work better in this instance but the authors should not be penalized for reporting this. I think a paper on a probabilistic method should at some point evaluate log likelihood in a case where the test distribution is the same as the training distribution. This complements experiments where there is dataset shift and we wish to show robustness. I would be very interested to know how useful the implied marginal likelihoods of the approximation where, as suggested for further work in the conclusion.

---

> ### Author Response · Authors · 2018-01-05
> **Response to Reviewer 1**
>
> Thank you very much for your thoughts and suggestions.
>
>
> >Given the complexity of the method, I think it would have helped to submit the code in anonymized form at this point.
>
> We will make the code available after the review period for ICLR. It is unfortunately spread out across multiple repositories, which we haven't open sourced yet, in particular we have integrated the calculation of the curvature factors that would also be needed for KFAC/KFRA into a currently internal version of Lasagne, so it would have been tricky to ensure that everything is fully anonymised.
>
>
> > There are also some experiments not there that would improve the contribution. Figure 1 should include a comparison to Hamiltonian Monte Carlo and the full Laplace approximation
>
> Thank you for pointing this out, we have added the corresponding figures to the manuscript and expanded the section. We have moved the figures of the unregularised Laplace approximations into the appendix and put figures for the regularised one into the main text, as they give a better fit to the HMC posterior.
>
>
> >It would be interesting (but perhaps not essential) to compare the Laplace approximation to other scalable methods from the literature such as that of Louizos and Welling 2016 which uses also used matrix normal distributions.
>
> We have added a comparison to a fully factorised Gaussian approximation as in Graves (2011) and Blundell et al. (2015) as this was also suggested by Reviewer 2. We attempted to train a network with an approximate matrix normal posterior as in Louizos & Welling (2016) by parameterising the Cholesky factors of the two covariance matrices, as this would most closely correspond to how the posterior is approximated by our Laplace approximation. However, this lead to poor classification accuracies and the authors confirmed that this approach wasn't successful for them either. They stated that instead the pseudo data ideas from the GP literature were crucial for the success of their method.

---

> > ### Comment · AnonReviewer3 · 2018-01-12
> > **Responses to rebuttal**
> >
> > Thank you for the new figures which along with those in the appendix clarify the significant role of regularization in the empirical use of the approximation.
> >
> > Thank you also for your discussion of the Louizos and Welling approximation - the comparison to fully factored Gaussian instead is informative.
> >
> > Overall I will be keeping my score the same.

---

### Decision · Program_Chairs · 2018-01-29
**ICLR 2018 Conference Acceptance Decision**

**Decision:**

Accept (Poster)

**Comment:**

This paper gives a scalable Laplace approximation which makes use of recently proposed Kronecker-factored approximations to the Gauss-Newton matrix. The approach seems sound and useful. While it is a rather natural extension of existing methods, it is well executed, and the ideas seem worth putting out there.